# MCNet: Monotonic Calibration Networks for Expressive Uncertainty Calibration in Online Advertising

Author
Institution
email@xxx.xx

## ABSTRACT

In online advertising, uncertainty calibration aims to adjust a ranking model's probability predictions to better approximate the true likelihood of an event, e.g., a click or a conversion. However, existing calibration approaches may lack the ability to effectively model complex nonlinear relations, consider context features, and achieve balanced performance across different data subsets. To tackle these challenges, we introduce a novel model called Monotonic Calibration Networks, featuring three key designs: a monotonic calibration function (MCF), an order-preserving regularizer, and a field-balance regularizer. The nonlinear MCF is capable of naturally modeling and universally approximating the intricate relations between uncalibrated predictions and the posterior probabilities, thus being much more expressive than existing methods. MCF can also integrate context features using a flexible model architecture, thereby achieving context awareness. The order-preserving and field-balance regularizers promote the monotonic relationship between adjacent bins and the balanced calibration performance on data subsets, respectively. Experimental results on both public and industrial datasets demonstrate the superior performance of our method in generating well-calibrated probability predictions.

## CCS CONCEPTS

• **Information systems** → **Computational advertising**.

## KEYWORDS

Uncertainty calibration, Online advertising, Monotonic networks

**ACM Reference Format:**
Author. 2025. MCNet: Monotonic Calibration Networks for Expressive Uncertainty Calibration in Online Advertising. In *Proceedings of The Web Conference 2025 (WWW '25)*. ACM, New York, NY, USA, 12 pages. https://doi.org/XXXXXXX.XXXXXXX

## 1 INTRODUCTION

In recent years, machine learning models have been extensively used to assist or even replace humans in making complex decisions in practical scenarios. Numerous applications, such as medical diagnosis [2], autonomous driving [1], and online advertising [11], have significant implications for people's safety or companies' economic

incomes. Therefore, the performance of these models is of paramount importance. Apart from classification accuracy or ranking performance, it is also crucial for the predicted score to accurately reflect the true likelihood of an event [8]. However, modern neural networks often struggle to predict accurate uncertainty, despite excelling at classification or ranking tasks. This limitation, influenced by their model architectures and regularization methods such as weight decay and batch normalization [8, 20], significantly hinders their application in real-world scenarios.

In this paper, we focus on the scenario of online advertising. Typically, ECPM (Effective Cost Per Mille) serves as a key metric for measuring the revenue earned by the platform for every 1,000 ad impressions. In Pay-Per-Click advertising, ECPM is estimated as the product of the predicted Click-Through Rate (CTR) and the bidding price, i.e., $CTR \times bid \times 1000$, which is used for ad ranking. This requires a model to output the predicted CTR that precisely reflects the probability of a user clicking on a given advertisement, as it directly influences bidding results and, consequently, the platform's revenue. However, as highlighted in [4, 16], the widely used deep ranking models heavily suffer from miscalibration, meaning the predicted CTR does not accurately represent the true probability.

To tackle this challenge, uncertainty calibration has been widely studied [4, 16, 25]. The goal is to train a well-calibrated predictor that can generate predictive scores accurately reflecting the actual probability of an event [5]. In this paper, we focus on the post-hoc calibration paradigm [4, 28] due to its flexibility and widespread adoption in practice. Post-hoc methods fix the base predictor and learn a new calibration function to transform the predicted scores from the base predictor into calibrated probabilities. As a result, they can be conveniently used as a model-agnostic, plug-and-play module placed on top of the base predictor in real systems.

Post-hoc methods mainly contain binning-based methods [28, 29], scaling-based methods [13, 17], and hybrid methods [4, 16]. Binning-based methods, such as Histogram Binning [28] and Isotonic Regression [29], first divide data samples into multiple bins and then directly use the empirical posterior probability as the calibrated probability for each bin. Consequently, their calibration function is a piecewise constant function (see Fig. 1a), making the samples within a bin lose their order information. To alleviate this, scaling-based methods employ parametric functions for calibration. While preserving the order information among samples, they still face limitations in expressiveness due to their strong assumptions on the distributions of the class-conditional predicted scores, such as the Gaussian distribution in Platt Scaling [17]. Moreover, hybrid methods integrate binning-based and scaling-based methods into a unified solution to fully leverage their advantages. Representative methods, including SIR [4], NeuCalib [16], and AdaCalib [24], typically learn a piecewise linear calibration function (see Fig. 1 (b) and (c)). Therefore, they are incapable of learning a perfect

 

calibration function when dealing with complex nonlinear relationships between uncalibrated scores and the true data distribution, which is often encountered in real-world applications. We defer to Section 4.1 for a more detailed and intuitive analysis.

In addition to limited expressiveness, existing post-hoc methods also struggle to adaptively capture varying miscalibration issues across contexts, and fail to achieve a balanced performance across different data fields. Specifically, NeuCalib introduces an additional module to capture context information, which heavily relies on an additional dataset. On the other hand, AdaCalib learns independent calibration functions for different data subsets, but it suffers from the data-efficiency issue. *In summary, existing methods fall short in developing an effective calibration function due to several key issues: their limited expressiveness, lack of context-awareness, and failure to consider field-balance calibration performance.*

In this paper, we propose a novel hybrid approach, **M**onotonic **C**alibration **Net**work (**MCNet**), designed to address the aforementioned challenges in uncertainty calibration. Like other hybrid methods, MCNet comprises a binning phase for dividing samples into multiple bins and a scaling phase for learning the calibration function for each bin. The success of MCNet hinges on three key designs, including a monotonic calibration function (MCF), an order-preserving regularizer, and a field-balance regularizer. Firstly, the MCF serves as a powerful approximator for capturing the intricate relationship between uncalibrated scores and the true data distribution (see Fig. 1 (d)). It is constructed using monotonic neural networks, ensuring the monotonically increasing property by enforcing the positivity of its derivative. Additionally, MCF can effectively model uncalibrated scores and context features with a flexible model architecture, thus achieving context-awareness efficiently. Secondly, the order-preserving regularizer is intended to promote monotonicity between different bins by penalizing calibration functions of two adjacent bins that violate the relative order at the split point of the two bins. The proposed calibration function, along with this regularizer, effectively address the first two issues. Additionally, we introduce a field-balance regularizer to address the third issue, which penalizes the variance of the calibration performance across different fields. By incorporating this regularizer, MCNet can attain a more balanced calibration performance.

In summary, this paper makes the following contributions:

- We propose a novel hybrid approach, Monotonic Calibration Networks, to achieve expressive, monotonically increasing, context-aware, and field-balanced uncertainty calibration.
- We design a monotonic calibration function, constructed using monotonic neural networks, to capture the complex relationship between uncalibrated scores and the true data distribution.
- We propose an order-preserving regularizer and a field-balance regularizer, which can significantly preserve order information and effectively promote balanced calibration performance among different fields, respectively.
- We conduct extensive experiments on two large-scale datasets: a public dataset AliExpress and a private industrial dataset from the advertising platform of Huawei browser, encompassing both click-through rate and conversion rate prediction tasks. The empirical results clearly demonstrate the effectiveness of MCNet in generating well-calibrated predictions.

## 2 RELATED WORK

### 2.1 Uncertainty Calibration

In real-world scenarios, the learned model must handle data samples from diverse contexts with varying data distributions. Therefore, it is crucial for the calibration function to consider the contextual information to enable adaptive calibration across contexts. Here, we broadly categorize existing uncertainty calibration approaches into two types: context-agnostic calibration [4, 8, 14, 17, 28] and context-aware calibration [16, 24].

**Context-agnostic calibration.** The uncalibrated score is the unique input of the calibration function. The binning-based methods, such as Histogram Binning [28] and Isotonic Regression [29], divide the samples into multiple bins, according to the sorted uncalibrated probabilities. In these non-parametric methods, the calibrated probability within each bin is the bin's posterior probability. Isotonic Regression merges adjacent bins to ensure the bins' posterior probabilities are non-decreasing. The scaling-based approaches, such as Platt Scaling [17] and Gamma Scaling [13], propose parametric functions that map the uncalibrated scores to calibrated ones. These parametric functions assume that the class-conditional scores follow Gaussian distribution (Platt Scaling) or Gamma distribution (Gamma Scaling). Smoothed Isotonic Regression (SIR) [4] learns a monotonically increasing calibration function with isotonic regression and linear interpolation, thus jointly exploiting the strengths of the binning- and scaling-based methods.

**Context-aware calibration.** In addition to the uncalibrated score, the context information, such as the field id denoting the source of data samples, has also been considered recently. NeuCalib, as the pioneering work, uses a univariate calibration function to transform the uncalibrated logits, and an auxiliary neural network that considers the sample features for context-adaptive calibration. However, it is important to note that the calibration function itself in NeuCalib does not directly consider the context information. AdaCalib [24], on the other hand, divides the validation set into several fields and learns an isotonic calibration function for each field using the field's posterior statistics. This approach, however, suffers from data efficiency issues due to the need for field-specific calibration. Both NeuCalib and AdaCalib adopt piecewise linear calibration functions, which may lack expressiveness when modeling the complex nonlinear relationships between the uncalibrated scores and the true data distribution. Our approach, MCNet, addresses this issue by learning nonlinear calibration functions with monotonic neural networks. Additionally, MCNet can naturally achieve context awareness by incorporating the context feature as input to its monotonic calibration function. Moreover, MCNet is equipped with a novel field-balance regularizer to ensure more balanced calibration performance across various fields.

### 2.2 Monotonic Neural Networks

Monotonic neural networks are models that exhibit monotonicity with respect to some or all inputs [3, 18]. Pioneering methods like MIN-MAX networks [19] achieve monotonicity through monotonic linear embeddings and max-min-pooling. Daniels and Velikova [3] extended this approach to construct partially monotone networks that are monotonic with respect to a subset of inputs. However,

these methods can be challenging to train, which limits their practical adoption. Further, deep lattice networks (DLNs) [27] is designed to combine linear embeddings, lattices, and piecewise linear functions to build partially monotone models. Other recent work, such as UMNN [23] and LMN [15], ensures monotonicity by learning a function whose derivative is strictly positive. Our work is inspired by monotonic neural networks, and is generally applicable to any implementation of monotonic neural networks.

## 3 PRELIMINARIES

### 3.1 Problem Formulation

In this paper, we study the problem of uncertainty calibration and formulate it from the perspective of binary classification. In binary classification, the aim is to predict the label $y \in \{0, 1\}$ of a data sample given its feature vector $\boldsymbol{x} \in \mathcal{X}$ by learning a predictor $g(\cdot)$ with a labeled training dataset. Then, given a data sample $\boldsymbol{x}$, the predicted probability of positive label is $\hat{p} = g(\boldsymbol{x})$, where the positive label refers to 1 and the negative label refers to 0.

Currently, the most widely used predictors such as logistic regression and deep neural networks are not well calibrated [4, 11]. It means that the predicted probability $\hat{p}$ could not accurately represent the true probability of the event $E[Y|X]$ defined as

$$E[Y|X = \boldsymbol{x}] = \lim_{\epsilon \to 0^+} P(Y = 1 | \|X - \boldsymbol{x}\| \leq \epsilon). \quad (1)$$

To tackle this problem, the post-hoc calibration, as a common paradigm, fixes the base predictor and learns a new mapping function to transform the raw model output $\hat{p}$ into the calibrated probability. Specifically, the aim of uncertainty calibration in post-hoc calibration is to find a function $f^*$ that takes the predicted score from $g$ as input such that the calibration error could be minimized, i.e.,

$$f^* = \arg\min_f \int_{\mathcal{X}} (E[Y|X = \boldsymbol{x}] - f(g(\boldsymbol{x})))^2 d\boldsymbol{x}. \quad (2)$$

In practice, the calibration function $f$ is learned based on a validation dataset $\mathcal{D}_{\text{val}} = \{(\boldsymbol{x}^{(i)}, y^{(i)})\}_{i=1}^N$ with $N$ samples. It can be a parametric or non-parametric function.

To evaluate the calibration performance, Eq. (2) is not a feasible metric due to the unobservable event likelihood $E[Y|X = \boldsymbol{x}]$. A common practice is to utilize the empirical data to approximate the true likelihood and quantify the calibration performance. Many metrics have been proposed. Predicted click over click (PCOC) [7, 9], as the most commonly used metric, calculates the ratios of the average calibrated probability and the posterior probability as

$$\text{PCOC} = (\frac{1}{|\mathcal{D}|} \sum_{i=1}^{|\mathcal{D}|} p^{(i)}) / (\frac{1}{|\mathcal{D}|} \sum_{i=1}^{|\mathcal{D}|} y^{(i)}), \quad (3)$$

where $\mathcal{D} = \{(\boldsymbol{x}^{(i)}, y^{(i)})\}_{i=1}^{|\mathcal{D}|}$ is the test dataset. PCOC is insufficient to evaluate the calibration performance, since it neither considers the distribution of calibrated probabilities nor takes into account the field information. To improve it, many more fine-grained metrics have been proposed. For example, calibration-$N$ [4] and probability-level calibration error (Prob-ECE) [14] make use of the calibrated distribution based on binning method. Further, a more reliable metric, i.e., field-level relative calibration error (F-RCE) is proposed [16], which is a weighted sum of the average bias of predictions in each

data subset divided by the true outcomes as

$$\text{F-RCE} = \frac{1}{|\mathcal{D}|} \sum_{c=1}^{|C|} N_c \frac{|\sum_{i=1}^{|\mathcal{D}|} (y^{(i)} - p^{(i)}) \mathbb{I}_c(c^{(i)})|}{\sum_{i=1}^{|\mathcal{D}|} (y^{(i)} + \epsilon) \mathbb{I}_c(c^{(i)})}, \quad (4)$$

where $c$ represents a specific field feature (which is usually a part of feature vector $\boldsymbol{x}$), $N_c$ is the number of samples of field $c$ with $\sum_{c=1}^{|C|} N_c = |\mathcal{D}|$, $\epsilon$ is a small positive number (e.g., $\epsilon = 0.01$) to avoid division by zero, and $\mathbb{I}_c(\cdot)$ is an indicator function with value as 1 if the input is $c$ otherwise 0.

### 3.2 Key Properties of Calibration

To learn a well-performed calibration function, several key characteristics should be carefully considered and balanced.

**Expressiveness**. The underlying data distribution in real scenarios can be highly complex, and the discrepancy between this distribution and the learned base predictor $g$ can be substantial. Therefore, the mapping function $f$ must be sufficiently expressive to facilitate the complex nonlinear transformations required to calibrate uncalibrated scores to the true data distribution.

**Order-Preserving**. This suggests that the calibrated probability output by $f$ should preserve the order of the original scores produced by the uncalibrated model $g$. Typically, the base predictor $g$ is a strong deep neural network that excels in ranking tasks. For instance, sophisticated deep models are widely used in CTR prediction in industry [30]. This property allows us to improve the predicted probability while maintaining the ranking performance.

**Context-Awareness**. In many applications, the trained model needs to handle data samples from various contexts (e.g., domains or categories) with significantly different distributions. The calibration function $f$ should incorporate context information to achieve adaptive calibration. However, this property can conflict with the order-preserving property. Specifically, for samples in different contexts, ground-truth probabilities can differ despite the same uncalibrated scores due to different miscalibration issues, thus violating the order-preserving property. Consequently, careful balancing of these two properties is essential.

**Field-Balance**. It is crucial for the calibration model to perform consistently across different fields (e.g., domains or categories). Inconsistent performance can lead to various issues. For example, in an online advertising platform, if the calibration model $f$ overestimates the probabilities for samples from certain fields while underestimating them for others, it can result in overexposure in some fields and underexposure in others. This imbalance can cause unfairness and negatively impact the ad ecosystem.

## 4 OUR APPROACH

In this section, we propose a novel hybrid approach, **M**onotonic **C**alibration **Net**works (**MCNet**), for uncertainty calibration. We begin by motivating the proposal of MCNet through an analysis of existing methods, followed by a detailed description of our method.

### 4.1 Analysis and Motivation

To begin with, we provide a discussion of some representative methods in post-hoc paradigm in terms of the key properties. We also intuitively demonstrates the calibration function of different

 

methods with a toy example in Figure 1.

**Binning-based methods.** The binning-based methods, such as Histogram Binning [28], first divide the samples into multiple bins according to the sorted uncalibrated probabilities (in ascending order), and then obtain the calibrated probability within each bin by computing the bin's posterior probability. Suppose the samples are divided into $K$ bins as $\{[b_0, b_1), \cdots, [b_{k-1}, b_k), \cdots, [b_{K-1}, b_K)\}$. Then, the calibration function can be formulated as

$$f(g(\boldsymbol{x})) = \sum_{k=1}^{K} \frac{\sum_{i=1}^{N} y^{(i)} \cdot \mathbb{I}_{[b_{k-1}, b_k)}\left(g(\boldsymbol{x}^{(i)})\right)}{\sum_{i=1}^{N} \mathbb{I}_{[b_{k-1}, b_k)}\left(g(\boldsymbol{x}^{(i)})\right)} \mathbb{I}_{[b_{k-1}, b_k)}\left(g(\boldsymbol{x})\right), \quad (5)$$

where $\mathbb{I}_{[b_{k-1}, b_k)}(\cdot)$ is an indicator with value as 1 if the input falls into $[b_{k-1}, b_k)$ otherwise 0. Essentially, the calibration function is a piecewise constant function (see Fig. 1a). It gives the same calibrated probability to all samples of the same bin, and thus loses the ranking information. Besides, they could not achieve context-awareness, and do not have any mechanism to enhance field-balance.

**Hybrid methods.** The hybrid methods integrate the binning and scaling methods into a unified solution so as to make full use of their advantages, which have achieved state-of-the-art performance. Representative approaches, such as SIR [4], NeuCalib [16], and AdaCalib [24], have piecewise linear calibration functions (see Fig. 1b and 1c). They first obtain multiple bins similarly as the binning-based methods, and then learn a linear calibration function for each bin. The calibration function can be formalized as

$$f(g(\boldsymbol{x})) = \sum_{k=1}^{K} \left[ a_{k-1} + \left(g(\boldsymbol{x}) - b_{k-1}\right) \frac{a_k - a_{k-1}}{b_k - b_{k-1}} \right] \mathbb{I}_{[b_{k-1}, b_k)}\left(g(\boldsymbol{x})\right), \quad (6)$$

where $\{b_k\}_{k=0}^{K+1}$ are binning boundaries and $\{a_k\}_{k=0}^{K+1}$ are learned differently for hybrid models. Specifically, SIR directly calculates $\{a_k\}_{k=0}^{K+1}$ from the statistics of the validation dataset, NeuCalib denotes them as learnable model parameters, and AdaCalib applies neural networks to learn them, respectively. Although these existing methods gradually use more and more complicated functions to learn $\{a_k\}_{k=0}^{K+1}$, their calibration functions are essentially linear and have limited expressiveness. Besides, NeuCalib and AdaCalib rely on similar additional order-preserving constraints to keep the order information, and do not consider field-balance.

**Motivation.** Figure 1 provides an intuitive illustration of the calibration errors of existing methods. As we can see, these methods, because of their limited expressiveness, are unable to achieve perfect uncertainty calibration when confronted with complex non-linear transformation relations. This motivates us to design a more expressive calibration function with stronger modeling capabilities.

## 4.2 Monotonic Calibration Networks

As motivated above, we propose an expressive Monotonic Calibration Network that has the capacity to learn a perfect uncertainty calibration function, as shown in Figure 1d. MCNet is a hybrid method consisting of two phases, i.e., the binning phase and the scaling phase. In the binning phase, the validation samples in $\mathcal{D}_{val}$ are divided into $K$ bins with equal frequency, and the interval for the $k$-th bin is $[b_{k-1}, b_k)$. $b_0$ and $b_K$ are two pre-defined numbers to ensure that all samples can be assigned to a specific bin. In the scaling phase, $K$ calibration functions are designed and learned for the $K$ bins, respectively.

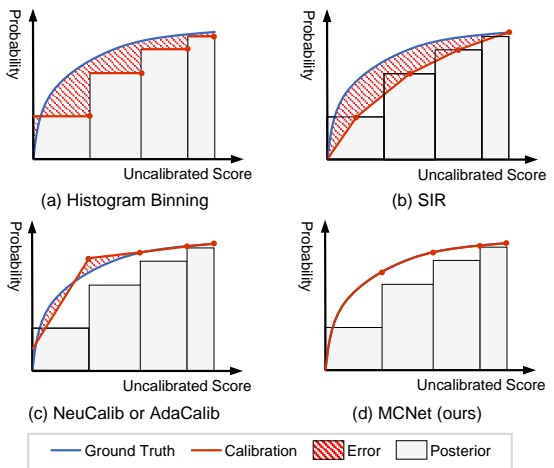

(a) Histogram Binning  (b) SIR

(c) NeuCalib or AdaCalib  (d) MCNet (ours)

Ground Truth — Calibration — Error — Posterior

**Figure 1: Illustration of different calibration functions.**

Figure 2 shows the architecture of our proposed Monotonic Calibration Network, which aims to achieve accurate calibration. MC-Net relies on three key designs: a monotonic calibration function, an order-preserving regularizer, and a field-balance regularizer. Next, we provide a detailed introduction to each of these components.

*4.2.1 Monotonic Calibration Function.* The monotonic calibration function (MCF) is built upon monotonic neural networks (MNNs) [3, 15, 23], which inherently possess the property of monotonicity and serve as strong approximators of the true data distribution. In this work, we implement the MCF based on an unconstrained MNN [23], which is designed with an architecture that ensures its derivative is strictly positive, thereby achieving the desired monotonicity.

Denote a sample as $(\boldsymbol{x}, c, y)$ where $\boldsymbol{x}$ represents all features and $c$ ($c \in C = \{c_1, c_2, \cdots, c_{|C|}\}$) represents a specific field feature (which is usually a part of $\boldsymbol{x}$). Here, the field feature $c$ is used as the context feature without loss of generality, and can be readily replaced with any other features. The proposed monotonic calibration function can jointly model the uncalibrated scores and context features with a flexible model architecture, thus achieving context-awareness. Compared with AdaCalib, which learns the calibration function for different fields independently, the MCF is shared by all fields within the same bin, making it more data-efficient.

Specifically, the calibration function MCF of MCNet for the $k$-th bin is formulated as follows:

$$f^k(g(\boldsymbol{x}), c) = \underbrace{\int_0^{g(\boldsymbol{x})} f_1^k\left(t, h^k(c; \Phi^k); \Theta_1^k\right) dt}_{\text{Integration Term}} + \underbrace{f_2^k\left(h^k(c; \Phi^k); \Theta_2^k\right)}_{\text{Bias Term}}, \quad (7)$$

where $f_1^k(\cdot; \Theta_1^k)$, $f_2^k(\cdot; \Theta_2^k)$, and $h^k(\cdot; \Phi^k)$ are parametric functions, and $\Theta_1^k$, $\Theta_2^k$, and $\Phi^k$ are the corresponding model parameters. $h^k(\cdot; \Phi^k)$ is an embedding function that transforms the input context feature id into embedding vectors. Both $f_1^k$ and $f_2^k$ can be implemented with any neural networks.

The MCF consists of an integration term and a bias term. Most importantly, the integration term jointly considers the uncalibrated score $g(\boldsymbol{x})$ and the context feature $c$ in a natural way to achieve both monotonicity and context-awareness. Figure 2 illustrates this term within the dashed box on the right side and demonstrates its

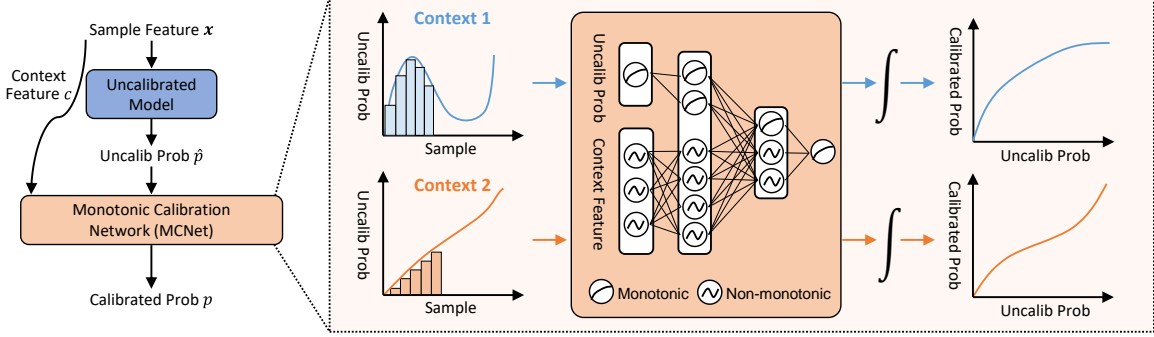

**Figure 2: Model architecture of MCNet. MCNet jointly models the uncalibrated score and the context feature to learn a monotonic calibration function. Given a specific context feature (e.g., context 1 and 2), MCNet generates the calibrated probabilities that are context-adaptive and monotonically increasing with the corresponding uncalibrated probabilities.**

properties intuitively. Specifically, $f_1^k$ is the derivative of $f^k$ with respect to the input $g(\boldsymbol{x})$, and it is designed to be a strictly positive parametric function, thus ensuring the function is monotonically increasing given the same context and contextually adaptive across different contexts. In experiments, we leverage the sigmoid as the activation function for $f_1^k$ to ensure the positiveness of its outputs. The bias term $f_2^k$ is designed to further capture the contextual information, and does not impact the monotonicity with respect to $g(\boldsymbol{x})$. It should be noted that the context feature $c$ is an optional input depending on whether contextual information plays an important role in the application.

Then, the overall calibration function of $K$ bins is formulated as

$$f\big(g(\boldsymbol{x}), c\big) = \sum_{k=1}^{K} f^k\big(g(\boldsymbol{x}), c\big) \mathbb{I}_{[b_{k-1}, b_k)}\big(g(\boldsymbol{x})\big), \quad (8)$$

where $\mathbb{I}_{[b_{k-1}, b_k)}(\cdot)$ is an indicator with value as 1 if the input falls into $[b_{k-1}, b_k)$ otherwise 0.

Theorem 1 (Expressiveness). *If the uncalibrated scores possess accurate order information and the ground truth calibration function is continuously differentiable, then the monotonic calibration function $f(\cdot)$ serves as a universal approximator of the ground truth function.*

This theorem suggests that MCNet is capable of learning the perfect nonlinear calibration function under certain conditions. We provide the proof of Theorem 1 in Appendix A.1.

To train MCNet, we quantify the instance-level calibration error with the negative log-likelihood loss, which is formulated as

$$\mathcal{L}_{logloss} = \frac{1}{N} \sum_{i=1}^{N} \Big[ -y^{(i)} \log p^{(i)} - (1 - y^{(i)}) \log(1 - p^{(i)}) \Big], \quad (9)$$

where $p^{(i)} = f\big(g(\boldsymbol{x}^{(i)}), c^{(i)}\big)$ is the calibrated probability obtained via the MCF.

*4.2.2 Order-Preserving Regularizer.* The proposed monotonic calibration function (Eq. (7)) can ensure the monotonicity within samples of the same bin, while samples across different bins are not constrained. To address this issue, we design a regularizer to encourage the monotonicity between different bins as follows

$$\mathcal{L}_{order} = \sum_{i=1}^{|C|} \sum_{k=1}^{K-1} \max \Big\{ f^k\big(g(b_k), c_i\big) - f^{k+1}\big(g(b_k), c_i\big), 0 \Big\}. \quad (10)$$

It penalizes calibration functions of the two adjacent bins that violate the monotonicity, i.e., the ending of the $k$-th calibration curve (which is located between $b_{k-1}$ and $b_k$) is greater than the beginning of the $(k+1)$-th one. With this regularizer, MCNet can preserve the order information of all samples if no context feature is given, i.e., the output of $h^k$ is set to an all-zero vector. If the field id is used as the context feature for calibration, MCNet can also maintain the order-preserving property within each field, while it might achieve better calibration performance for each specific field due to the consideration of context features.

*4.2.3 Field-Balance Regularizer.* In many real scenarios, it is highly important to keep a balance of the calibration performance among different fields. For example, in online advertising platform, a balanced performance can enhance fairness of different bidders and improve the healthiness of the business ecosystem. PCOC and its variants are widely leveraged to quantify the calibration performance [4]. However, it can easily cause high variance issue if directly used as the evaluation metric in our regularizer due to its division operation. To avoid this issue, we design a new metric for quantifying the calibration performance for field $c$ as follows

$$DIFF_c = \frac{1}{N} \sum_{i=1}^{N} \Big[ f\big(g(\boldsymbol{x}^{(i)}), c^{(i)}\big) - y^{(i)} \Big] \cdot \mathbb{I}_c\big(c^{(i)}\big), \quad (11)$$

where $\mathbb{I}_c(\cdot)$ is an indicator function with value as 1 if the input is $c$ otherwise 0. Different from PCOC, it computes the diference between the calibrated probability and the posterior probability of the data through subtraction, which can quantify both overestimation and underestimation as PCOC. To enhance the balance of calibration performance among different fields, we use the standard variance of DIFF of different fields as the field-balance regularizer

$$\mathcal{L}_{balance} = \sqrt{\frac{\sum_{i=1}^{|C|} (DIFF_i - \overline{DIFF})^2}{|C|}}, \quad (12)$$

where $\overline{DIFF}$ is the mean of $\{DIFF_i\}_{i=1}^{|C|}$.

The overall training loss for MCNet is formulated as

$$\mathcal{L}_{MCNet} = \mathcal{L}_{logloss} + \beta \cdot \mathcal{L}_{order} + \alpha \cdot \mathcal{L}_{balance}, \quad (13)$$

where $\beta$ and $\alpha$ are hyperparameters to control the importance of the regularization terms.

*4.2.4 Discussion.* MCNet flexibly balances the properties of order-preserving and context-awareness. Specifically, the calibration function $f_k$ is strictly monotonically increasing with respect to the input uncalibrated score, thereby perfectly preserving the order information for samples of the same context within the same bin. Further, the relative order of samples across bins can be easily constrained using the order-preserving regularizer. For samples of different contexts, the ground-truth probability can differ even with the same uncalibrated score, due to different miscalibration issues. Our MCNet enables context-adaptive calibration by naturally modeling the contextual information. For scenarios where context features convey limited information, MCNet can directly disregard them and preserve the order information across contexts.

## 4.3 Training Algorithm

The proposed MCNet cannot be trivially trained with stochastic gradient descent (SGD) methods due to the existence of integration operation in Eq. (7). To tackle this problem, we use Clenshaw-Curtis quadrature (CCQ) [6, 21] to compute the forward and backward integration. In CCQ, the integration is computed by constructing a polynomial approximation, involving $T$ forward operation of $f_1$. Thus, the complexity of MCNet is a constant times of a normal neural network positively depending on $T$. Luckily, the $T$ forward operations can be computed in parallel, making it time efficient. In the backward step, we integrate the gradient instead of computing the gradient of the integral to avoid storing additional results, making it memory efficient. Thus, CCQ enables the computation of gradients of MCNet efficiently and thus the optimization of it with SGD effectively. Appendix A.2 provides the detailed training algorithms based on CCQ as well as the empirical analysis of time- and memory-efficiency.

## 5 EXPERIMENTS

In this section, extensive experiments are conducted to investigate the following research questions (RQs).

- RQ1: How does MCNet perform on the calibration tasks compared with the state-of-the-art baseline approaches?
- RQ2: What are the effects of the auxiliary neural network and the field-balance regularizer on the performance of MCNet?
- RQ3: What are the strengths of nonlinear calibration functions learned by MCNet?
- RQ4: How will the hyperparameters affect MCNet's performance?

### 5.1 Experimental Setup

*5.1.1 Datasets.* The experiments are conducted on two large-scale datasets: one public dataset (AliExpress[1]) and one private industrial dataset (Huawei Browser). Both datasets are split into three subsets for training, validation, and testing, respectively. For AliExpress, the field feature $c$ is set as the country where the data are collected. With such a categorical feature as the field information, AliExpress can be divided into 4 fields (i.e., 4 disjoint subsets), representing the 4 countries. The Huawei Browser dataset is extracted directly from the Huawei online advertising system with samples across 9 days. It is partitioned into 3 fields, indicating the 3 advertisement

---

[1] https://tianchi.aliyun.com/dataset/74690

sources. More detailed descriptions are provided in Appendix B.1.

*5.1.2 Baselines.* We make comparisons with three categories of baselines. (1) Binning-based methods: Histogram Binning [28] and Isotonic Regression [29]. (2) Scaling-based methods: Platt Scaling [17], Gaussian Scaling [13], and Gamma Scaling [13]. (3) Hybrid methods: SIR [4], NeuCalib [16], and AdaCalib [24]. More detailed decriptions of them are provided in Appendix B.2.

Our method, **MCNet**, is also a hybrid approach, including two variants: **MCNet-None** and **MCNet-Field**. MCNet-None is a variant without considering the context feature. The output of embedding function $h^k$ is set to an all-zero vector. MCNet-Field is a variant that takes the field information as the context feature.

*5.1.3 Implementation Details.* In our methods, the parametric functions $f_1(\cdot)$ and $f_2(\cdot)$ for each bin are implemented as multilayer perceptions (MLPs) with two 128-dimensional hidden layers. $h(\cdot)$ is an embedding lookup table with an embedding dimension of 128. The balance coefficient $\beta$ in Eq. (13) is set as 1. The proposed calibration models are trained using the Adam optimizer [12] with a batch size of 2048. The default learning rates for MCNet-None and MCNet-Field are 1e-5 and 1e-4, respectively. The base predictor $g(\cdot)$ is deep click-through rate prediction model [22]. The baseline calibration approaches are implemented based on their papers. We set the number of bins to 20 for all methods that require binning. The base predictor is trained on the training set. All calibration methods are trained using the validation set and evaluated on the test set. We conduct both the CTR and CVR calibration on the AliExpress and Huawei Browser datasets. We choose PCOC as the field-agnostic calibration metric, F-RCE as the field-level calibration metric, and AUC score as the ranking metric.

### 5.2 Performance Study (RQ1)

As shown in Table 1, compared with the base predictor, all approaches have improved calibration metrics (i.e., PCOC and F-RCE) on both datasets. For example, on the AliExpress dataset, MCNet-None significantly reduces the F-RCE from 16.10% to 1.77% on the CTR task and from 34.21% to 10.07% on the CVR task. In comparison with the baseline calibration approaches, MCNet-None yields the best PCOC and F-RCE on both the CTR and CVR tasks based on results of the paired-t-test compared with the best baseline, indicating MCNet-None is more expressive to approximate the posterior probabilities of the test set. On the Huawei Browser dataset, MCNet-None still obtains competitive calibration performance on both tasks. With the field information incorporated, MCNet-Field achieves the lowest F-RCE scores on both CTR and CVR tasks and the second-best PCOC on the CTR task. Thus, MCNet-None and MCNet-Field are more favorable on AliExpress and Huawei Browser, respectively, and they can be chosen based on their performance on a specific task. Since MCNet-Field incorporates the field features, priority can be given to MCNet-Field if the field features are vital. Moreover, on these two datasets, both MCNet-None and MCNet-Field maintain AUC scores the same as or close to those of the base predictor, thus largely preserving the original ranking of uncalibrated probabilities. Therefore, our methods, MCNet-None and MCNet-Field, excel in uncertainty calibration by learning nonlinear calibration functions with monotonic neural networks.

**Table 1: Results on the AliExpress and Huawei Browser datasets. The highest results in each column are in boldface, the second-best values are underlined, and the values inside "()" represent the standard deviation across three different runs.**

| Method | CTR (AliExpress) | | | CVR (AliExpress) | | | CTR (Huawei Browser) | | | CVR (Huawei Browser) | | |
|---|---|---|---|---|---|---|---|---|---|---|---|---|
| | PCOC | F-RCE | AUC | PCOC | F-RCE | AUC | PCOC | F-RCE | AUC | PCOC | F-RCE | AUC |
| Base | 0.7727 | 16.10% | 0.7216 | 1.4324 | 34.21% | **0.7892** | 1.0750 | 3.18% | **0.8758** | 0.9814 | 1.60% | **0.8500** |
| Histogram Binning | 0.8467 | 10.86% | 0.7198 | 1.1966 | 16.04% | 0.7865 | 0.9634 | 2.15% | 0.8687 | 1.0042 | 0.80% | 0.8493 |
| Isotonic Regression | 0.8482 | 10.74% | 0.7215 | 1.1989 | 16.18% | 0.7883 | 0.9701 | 0.76% | **0.8758** | 1.0037 | 0.81% | 0.8499 |
| Platt Scaling | 0.8441 | 11.03% | 0.7216 | 1.3700 | 29.39% | **0.7892** | 0.9692 | **0.83%** | **0.8758** | **1.0030** | 0.95% | **0.8500** |
| | (0.0050) | (0.35%) | (0.0000) | (0.0024) | (0.18%) | (0.0000) | (0.0069) | (0.17%) | (0.0000) | (0.0006) | (0.05%) | (0.0000) |
| Gaussian Scaling | 0.8440 | 11.05% | 0.7216 | 1.3092 | 24.75% | **0.7892** | 0.9638 | 1.11% | **0.8758** | 1.0045 | 0.93% | **0.8500** |
| | (0.0101) | (0.72%) | (0.0000) | (0.0084) | (0.66%) | (0.0000) | (0.0201) | (0.49%) | (0.0000) | (0.0023) | (0.12%) | (0.0000) |
| Gamma Scaling | 0.8484 | 10.73% | 0.7216 | 1.2720 | 21.77% | **0.7892** | 0.9670 | 1.10% | **0.8758** | 1.0064 | 1.13% | **0.8500** |
| | (0.0070) | (0.50%) | (0.0000) | (0.0043) | (0.34%) | (0.0000) | (0.0238) | (0.55%) | (0.0000) | (0.0039) | (0.20%) | (0.0000) |
| SIR | 0.7821 | 15.44% | 0.7216 | 1.1913 | 15.60% | **0.7892** | 0.9448 | 1.51% | **0.8758** | 1.0037 | 0.82% | **0.8500** |
| NeuCalib | 0.8472 | 10.82% | 0.7216 | 1.1798 | 14.66% | **0.7892** | **0.9902** | 1.27% | **0.8758** | 1.0050 | 0.87% | **0.8500** |
| | (0.0040) | (0.29%) | (0.0000) | (0.0231) | (1.86%) | (0.0000) | (0.0052) | (0.05%) | (0.0000) | (0.0028) | (0.17%) | (0.0000) |
| AdaCalib | 0.8599 | 9.93% | **0.7217** | 1.1892 | 15.21% | 0.7880 | 0.9746 | 1.14% | 0.8757 | 1.0071 | 1.04% | 0.8499 |
| | (0.0274) | (1.94%) | (0.0000) | (0.0047) | (0.39%) | (0.0001) | (0.0367) | (0.95%) | (0.0000) | (0.0096) | (0.60%) | (0.0000) |
| MCNet-None (ours) | **0.9745** | **1.77%** | 0.7215 | **1.1094** | **10.07%** | **0.7892** | 0.9721 | 0.92% | **0.8758** | 1.0059 | 0.96% | 0.8499 |
| | (0.0141) | (1.00%) | (0.0001) | (0.0036) | (0.24%) | (0.0000) | (0.0021) | (0.00%) | (0.0000) | (0.0074) | (0.52%) | (0.0002) |
| MCNet-Field (ours) | 0.8642 | 9.63% | 0.7215 | 1.1728 | 14.03% | 0.7877 | 0.9804 | **0.83%** | 0.8757 | 1.0056 | **0.61%** | **0.8500** |
| | (0.0032) | (0.22%) | (0.0001) | (0.0048) | (0.29%) | (0.0001) | (0.0159) | (0.35%) | (0.0000) | (0.0012) | (0.10%) | (0.0000) |

## 5.3 Ablation Study (RQ2)

*5.3.1 Study on the auxiliary neural network.* An auxiliary neural network can be incorporated into MCNet as NeuCalib and AdaCalib, serving as an optional component to improve the ranking performance. The auxiliary neural network takes the sample feature vectors as inputs. When the auxiliary network is used, MCNet takes uncalibrated logit $\hat{l}$ as the input and outputs calibrated logit $l$. The calibrated probability is calculated by adding the outputs of the calibration function and the auxiliary neural network. An illustration is provided in Appendix A.3. Note that the auxiliary network relies on an independent validation set. If the validation set for calibration is a subset of the training set, the auxiliary network should be removed. Table 2 reports the calibration and ranking metrics with an auxiliary neural network (Aux) incorporated into each model, including AdaCalib, NeuCalib, MCNet-None, and MCNet-Field. The auxiliary neural network is implemented as a 2-layer MLP. The experimental settings are the same as those of Section 5.2.

By comparing the results in Table 2 and Table 1, it is observed that the auxiliary network increases the AUC scores of MCNet-None and MCNet-Field in most cases. For instance, on the CTR task of Huawei Browser dataset, the relative AUC improvements obtained by the auxiliary network are 1.10% for MCNet-None and 0.82% for MCNet-Field, respectively. Similar observations can be found in NeuCalib and AdaCalib. Hence, the auxiliary network can improve the ranking ability of a calibration model. However, the calibration metrics (i.e., PCOC and F-RCE) sometimes get worse with the auxiliary network integrated. For example, worse values of PCOC and F-RCE can be seen in MCNet-None-Aux compared with MCNet-None. It reveals that the auxiliary network may have a negative impact on a model's calibration ability.

*5.3.2 Study on the field-balance regularizer.* Table 3 shows the results with the field-balance regularizer (i.e., $\mathcal{L}_{balance}$, see Section 4.2.3) applied. A field's PCOC reveals the calibration performance on this field. It can be seen that the field-balance regularizer reduces the PCOC standard deviations of all calibration approaches. For example, the PCOC standard deviations of MCNet-None and MCNet-Field decrease from 3.56% to 2.93% and from 2.36% to 1.81%, respectively. Such observations demonstrate that the field-balance regularizer can improve the balance of calibration performance on different fields, thus promoting the fairness of the product ecosystem. In addition, under the field-balance regularizer, the overall PCOC is improved in most cases. Moreover, the field-balance regularizer achieves a reduced or comparable F-RCE, and maintains the AUC score. Thus, the field-balance regularizer can keep the calibration and ranking metrics while promoting the field-balance.

## 5.4 Calibration Function Analysis (RQ3)

Figure 3 shows the calibration functions of MCNet-None and three baselines, i.e., SIR, NeuCalib, and AdaCalib. These calibration functions are learned using the validation set. The blue bar is the posterior probability of test samples in each bin. The bin number is set as 10. As introduced in Section 4.1, the piecewise linear calibration function of SIR is constructed directly using the posterior probabilities of the validation set. Consequently, the two ends of each line are within the bins. The ordinate value of each endpoint is the bin's posterior probability. By comparing the calibration curve of SIR and the posterior statistics of the test set, it can be observed that the validation and test sets have distinct distributions of posterior probabilities. In comparison with the baseline approaches,

**Table 2: Ablation study on the auxiliary neural network.**

| Method | CTR (AliExpress) | | | CVR (AliExpress) | | | CTR (Huawei Browser) | | | CVR (Huawei Browser) | | |
|---|---|---|---|---|---|---|---|---|---|---|---|---|
| | PCOC | F-RCE | AUC | PCOC | F-RCE | AUC | PCOC | F-RCE | AUC | PCOC | F-RCE | AUC |
| Base | 0.7727 | 16.10% | 0.7216 | 1.4324 | 34.21% | 0.7892 | 1.0750 | 3.18% | 0.8758 | 0.9814 | 1.60% | 0.8500 |
| NeuCalib-Aux | 0.8966 | 7.33% | 0.7258 | 1.1175 | 9.58% | 0.7850 | 1.0334 | 1.24% | 0.8814 | 1.0060 | 0.62% | 0.8515 |
| AdaCalib-Aux | 0.8920 | 7.65% | 0.7244 | 0.9597 | 2.73% | 0.7873 | 1.0042 | 0.22% | 0.8847 | 1.0187 | 2.00% | 0.8519 |
| MCNet-None-Aux | 0.8857 | 8.10% | 0.7261 | 1.0168 | 2.58% | 0.7894 | 1.0273 | 1.19% | 0.8854 | 1.0061 | 1.40% | 0.8536 |
| MCNet-Field-Aux | 0.8908 | 7.73% | 0.7254 | 1.0122 | 1.94% | 0.7870 | 1.0002 | 0.20% | 0.8829 | 0.9949 | 0.60% | 0.8522 |

**Table 3: Ablation study on the field-balance regularizer with CTR task on AliExpress. "All" and "STD" denote the overall PCOC and the PCOC standard deviation of the four fields, respectively.**

| Model | $\mathcal{L}_b$ | F-RCE | AUC | PCOC | | | | | |
|---|---|---|---|---|---|---|---|---|---|
| | | | | All | Field 0 | Field 1 | Field 2 | Field 3 | STD |
| M-N | ✗ | 0.63% | 0.7214 | 0.991 | 0.997 | 0.915 | 0.942 | 0.930 | 3.56% |
| | ✓ | 0.72% | 0.7216 | 1.002 | 1.007 | 0.938 | 0.966 | 0.956 | 2.93% |
| M-F | ✗ | 9.56% | 0.7216 | 0.865 | 0.867 | 0.878 | 0.828 | 0.837 | 2.36% |
| | ✓ | 8.99% | 0.7213 | 0.873 | 0.875 | 0.864 | 0.849 | 0.833 | 1.81% |
| N | ✗ | 10.54% | 0.7216 | 0.851 | 0.855 | 0.803 | 0.822 | 0.798 | 2.59% |
| | ✓ | 10.02% | 0.7216 | 0.859 | 0.862 | 0.811 | 0.831 | 0.806 | 2.53% |
| A | ✗ | 7.51% | 0.7217 | 0.894 | 0.894 | 0.934 | 0.877 | 0.934 | 2.83% |
| | ✓ | 8.19% | 0.7216 | 0.885 | 0.885 | 0.891 | 0.874 | 0.853 | 1.67% |

\* M-N: MCNet-None, M-F: MCNet-Field, N: NeuCalib, A: AdaCalib, $\mathcal{L}_b$:$\mathcal{L}_{balance}$.

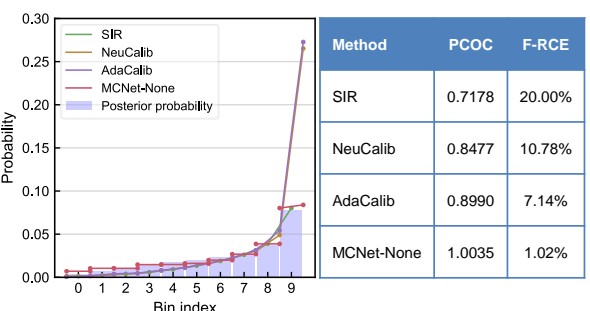

**Figure 3: Visualization of calibration functions.**

the calibration function of MCNet-None can more closely approximate the posterior probabilities of the test set. This point can also be supported by the preferable PCOC and F-RCE of MCNet-None. Note that, in MCNet-None, the gap between two adjacent calibration curves is caused by the order-preserving regularizer (see Section 4.2.2). With such an order-preserving regularizer, the calibrated probabilities are non-decreasing, thus keeping the original ranking of uncalibrated probabilities.

## 5.5 Hyperparameter Sensitivity Analysis (RQ4)

Figure 4 shows the calibration metrics of MCNet under two key hyperparameters, i.e., bin number and balance coefficient $\beta$. Bin number is the number of bins that the validation set is divided. In each bin, a monotonic calibration function is learned for calibration. $\beta$ is the balance coefficient in the overall loss function (i.e., Eq. (13)). The experiments are conducted on the AliExpress datasets with the F-RCE and PCOC of the CTR task reported. When investigating

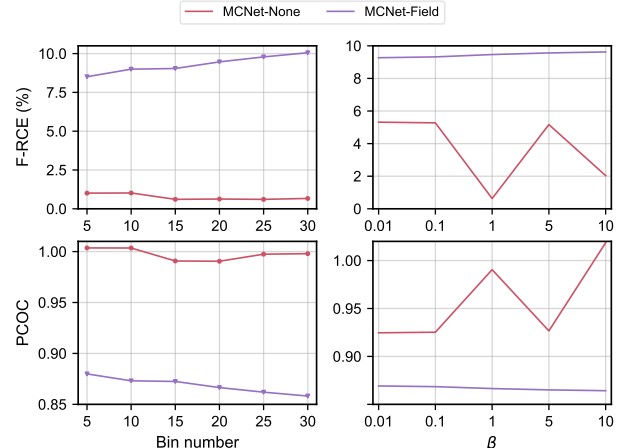

**Figure 4: Calibration metrics across different bin numbers and balance coefficients $\beta$.**

one hyperparameter, the remaining hyperparameters are set as the default values described in Section 5.1.3. Compared with MCNet-None, MCNet-Field is more sensitive to bin number while less sensitive to the balance coefficient $\beta$. When varying the bin number, the F-RCE and PCOC of MCNet-None only slightly change. The calibration metrics of MCNet-Field become worse under larger bin numbers. When changing the value of $\beta$, the calibration metrics of MCNet-Field remain stable, while those of MCNet-None fluctuate. The optimal setting of $\beta$ is 1.0 for MCNet-None. We also study the effect of training epochs, and provide an analysis on the robustness of MCNet against overfitting, in Appendix B.4.

## 6 CONCLUSION

We have proposed a novel hybrid method, Monotonic Calibration Networks (MCNet), to tackle the current challenges in uncertainty calibration for online advertising. MCNet is equipped with three key designs: a monotonic calibration function (MCF), an order-preserving regularizer, and a field-balance regularier. The proposed MCF is capable of learning complex nonlinear relations by leveraging expressive monotonic neural networks. Additionally, its flexible architecture enables efficient joint modeling of uncalibrated scores and context features, facilitating effective context-awareness. The two proposed regularizers further enhance MCNet by improving the monotonicity increasing property for preserving order information and the field-balanced calibration performance, respectively. Finally, extensive experiments on both public and industrial datasets are conducted to demonstrate the superiority of MCNet in the CTR and CVR tasks.

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

# Appendices

## Appendix A    MODEL DETAILS

### A.1    Proof of Theorem 1

THEOREM 1 (EXPRESSIVENESS). *If the uncalibrated scores possess accurate order information and the ground truth calibration function is continuously differentiable, then the monotonic calibration function $f(\cdot)$ serves as a universal approximator of the ground truth function.*

PROOF. If the uncalibrated scores possess accurate order information, then the true probability is monotonically increasing with respect to the uncalibrated scores. This implies that the ground truth calibration function has a strictly positive derivative. The derivative of the proposed calibration function $f^k(\hat{p}, c)$ is given by $\frac{d}{d\hat{p}} f^k(\hat{p}, c) = f_1^k(\hat{p}, h^k(c))$, which is strictly positive, as guaranteed by the output activation. Based on [10], a multi-layer feedforward network with sufficient hidden units is a universal approximator for our problem. Thus, the function $f_1^k(\cdot)$ can be implemented with multi-layer feedforward networks with sufficient hidden units, allowing it to readily accommodate any positive continuous functions. Hence, $f^k(\hat{p}, c)$ can precisely match the groundtruth calibration function of the $k$-th bin.                               □

### A.2    Algorithm and Complexity

Algorithm 1 and Algorithm 2 show the detailed training procedures for both forward and backward integration with Clenshaw-Curtis quadrature (CCQ) [21, 23], respectively. In CCQ, the integration is computed by constructing a polynomial approximation, involving $T$ (set to 50 empirically) forward operation of $f_1$. Thus, the complexity of MCNet is a constant times of a normal neural network positively depending on $T$. Luckily, the $T$ forward operations can be computed in parallel, making it time efficient. In the backward step, we integrate the gradient instead of computing the gradient of the integral to avoid storing additional results, making it memory efficient.

In addition, we provide the training time per epoch and the GPU memory consumption of closely-related baselines and our methods in Table 4, verifying that MCNet is both time- and memory-efficient. Although MCNet takes several times longer than the baseline methods, this is acceptable in practical applications because the calibration dataset is usually much smaller than the training dataset.

### A.3    Auxiliary Neural Network

Figure 5 illustrates the model architecture with an additional auxiliary network incorporated into the MCNet model. The inputs are the sample feature vectors (including the context features). The calibrated probability is obtained by adding the outputs of MCNet and the auxiliary network.

## Appendix B    MORE EXPERIMENTS

### B.1    Experimental Datasets

The experiments are conducted on two real-world datasets: one public dataset (AliExpress[2]) and one private industrial dataset (Huawei

---

[2] https://tianchi.aliyun.com/dataset/74690

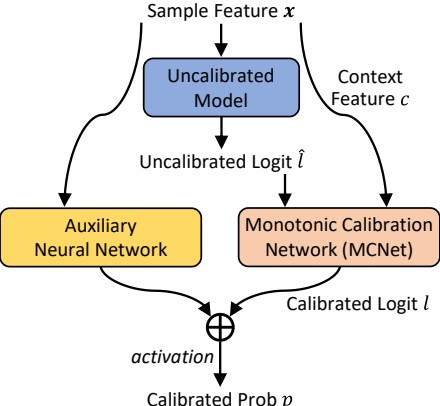

**Figure 5: Model architecture with an additional auxiliary network.**

**Table 4: Training time per epoch (min) and GPU memory consumption (MiB).**

| Method | N | AliExpress | | Huawei Browser | |
|---|---|---|---|---|---|
| | | Time | Memory | Time | Memory |
| NeuCalib | - | 0.63 | 1,457 | 23.0 | 23,855 |
| AdaCalib | - | 1.3 | 1,497 | 21.7 | 24,263 |
| MCNet-None | 10 | 5.5 | 1,515 | 69.6 | 24,889 |
| MCNet-None | 20 | 5.5 | 1,515 | 79.0 | 24,838 |
| MCNet-None | 50 | 5.4 | 1,509 | 74.6 | 24,824 |

Browser). We provide the detailed statistics of experimental datasets, i.e., AliExpress and Huawei Browser, in Table 5. For AliExpress, the provided training and test sets are split along the time sequence. Since the timestamp information is not available in the original training set, following AdaCalib [24], we split the original training set to be a new training set and a validation set with a proportion of 3:1. Field feature $c$ (see Section 4.2.1) is set as the country where the data are collected. With such a categorical feature as the field information, AliExpress can be divided into 4 fields (i.e., 4 disjoint subsets), representing the 4 countries. The Huawei Browser dataset is extracted directly from the Huawei online advertising system, which has samples across 9 days. To simulate the real scenario, we split this dataset by the date, i.e., the first 7 days for training, the 8th day for validation, and the 9th day for testing. Huawei Browser dataset is partitioned into 3 fields, indicating the 3 advertisement sources.

### B.2    Baseline Methods

We make comparisons with three categories of baselines. (1) **Binning-based methods**: **Histogram Binning** [28] partitions the ranked uncalibrated scores into multiple bins, and assigns the calibrated probability of each bin to be the bin's posterior probability. **Isotonic Regression** [29] improves over Histogram Binning by merging the adjacent bins to ensure the bin's posterior probability keeps

---

**Algorithm 1:** Forward Integration for MCNet with Clenshaw-Curtis quadrature

| | |
|---|---|
| **Input** | : $p$: The superior integration bounds, i.e., the uncalibrated score $g(x)$ in Eq. (7). |
| | $h$: The vector that representing embeddings of the input feature id, and $h$ is denoted as $h^k$ for the $k$-th bin. |
| **Output** | : $f$: The integral of $\int_0^p f_1(t; h; \Phi) dt$, and $f$ is denoted as $f^k$ for the $k$-th bin. |
| **Hyperparameters** | : $f_1$: A derivable function $\mathbb{R} \to \mathbb{R}$ with model parameters as $\Phi$. |
| | $T$: The number of integration steps. |

1 // Compute Clenshaw-Curtis weights and evaluation steps
2 $w, \delta_p = \text{COMPUTE\_CLENSHAW\_CURTIS\_WEIGHTS}(T)$
3 $f = 0$
4 **for** $i = 1, \ldots, T$ **do**
5     $p_i = p_0 + \frac{1}{2}(p - p_0)(\delta_p[i] + 1)$       // Compute the next point to evaluate
6     $\delta_f = f_1(p_i; h; \Phi)$
7     $f = f + w[i]\delta_f$
8 **end**
9 $f = \frac{f}{2}(p - p_0)$
10 **return** $f$

---

**Algorithm 2:** Backward Integration for MCNet with Clenshaw-Curtis quadrature

| | |
|---|---|
| **Input** | : $p$: The superior integration bounds. |
| | $h$: The vector that representing embeddings of the input feature id, and $h$ is denoted as $h^k$ for the $k$-th bin. |
| | $\nabla_{out}$: The derivatives of the loss function with respect to $\int_0^p f_1(t; h; \Phi) dt$ for all $p$. |
| **Output** | : $\nabla_\Phi$: The gradient of $\int_0^p f_1(t; h; \Phi) dt$ with respect to the parameters $\Phi$ of $f_1$. |
| | $\nabla_h$: The gradient of $\int_0^p f_1(t; h; \Phi) dt$ with respect to feature embeddings $h$. |
| **Hyperparameters** | : $f_1$: A derivable function $\mathbb{R} \to \mathbb{R}$ with model parameters as $\Phi$. |
| | $T$: The number of integration steps. |

1 // Compute Clenshaw-Curtis weights and evaluation steps
2 $w, \delta_p = \text{COMPUTE\_CLENSHAW\_CURTIS\_WEIGHTS}(T)$
3 $f, \nabla_\Phi, \nabla_h = 0, 0, 0$
4 **for** $i = 1, \ldots, T$ **do**
5     $p_i = p_0 + \frac{1}{2}(p - p_0)(\delta_p[i] + 1)$       // Compute the next point to evaluate
6     $\delta_F = f_1(p_i; h; \Phi)$
7     // Sum up for all samples of the batch the gradients with respect to inputs $h$
8     $\delta_{\nabla_h} = \sum_{j=1}^{B} \nabla_{h^j}\left(\delta_f^j\right)\nabla_{out}^j(p^j - p_0^j)$
9     // Sum up for all samples of the batch the gradients with respect to parameters $\Phi$
10     $\delta_{\nabla_\Phi} = \sum_{j=1}^{B} \nabla_\Phi\left(\delta_f^j\right)\nabla_{out}^j(p^j - p_0^j)$
11     $\nabla_h = \nabla_h + w[i]\delta_{\nabla_h}$
12     $\nabla_\Phi = \nabla_\Phi + w[i]\delta_{\nabla_\Phi}$
13 **end**
14 **return** $\nabla_\Phi, \nabla_h$

---

increasing. (2) **Scaling-based methods**: They design parametric calibration functions with the assumption that the class-conditional scores follow the Gaussian distribution (**Platt Scaling** [17] and **Gaussian Scaling** [13]) or Gamma distribution (**Gamma Scaling** [13]). (3) **Hybrid methods**: These methods borrow ideas from both the binning- and scaling-based methods. **SIR** [4] constructs calibration functions using isotonic regression and linear interpolation. **NeuCalib** [16] computes the calibrated probabilities with a linear calibration function and a field-aware auxiliary neural network. **AdaCalib** [24] learns one linear calibration function for each

field using the field's posterior statistics.

### B.3 Evaluation Metrics

To make the evaluations comprehensive, we provide the expected calibration error (ECE) scores of MCNet and three critical baselines (SIR, NeuCalib, and AdaCalib) on both AliExpress and Huawei Browser in Table 6. Fellow NeuCalib [16], ECE is calculated by

$$\text{ECE} = \frac{1}{|\mathcal{D}|}\sum_{k=1}^{K}\left|\sum_{i=1}^{|\mathcal{D}|}(y^{(i)} - p^{(i)})\mathbb{I}_{[b_{k-1}, b_k]}\left(p^{(i)}\right)\right|. \quad (14)$$

**Table 5: Statistics of the AliExpress and Huawei Browser datasets.**

|  | Field ID | Training #Impression | #Click | #Conversion | Validation #Impression | #Click | #Conversion | Test #Impression | #Click | #Conversion |
|---|---|---|---|---|---|---|---|---|---|---|
| AliExpress | 0 | 8,355,111 | 166,433 | 5,742 | 2,785,326 | 55,176 | 1,913 | 5,115,069 | 123,544 | 4,191 |
|  | 1 | 219,651 | 5,451 | 311 | 73,068 | 1,828 | 158 | 132,654 | 3,960 | 234 |
|  | 2 | 445,559 | 9,992 | 514 | 148,164 | 3,240 | 108 | 253,662 | 7,052 | 415 |
|  | 3 | 98,100 | 2,112 | 117 | 32,915 | 739 | 41 | 57,916 | 1,551 | 71 |
|  | All | 9,118,421 | 183,988 | 6,684 | 3,039,473 | 60,983 | 2,220 | 5,559,301 | 136,107 | 4,911 |
| Huawei Browser | 0 | 273,615,005 | 1,114,264 | 515,301 | 39,978,390 | 147,305 | 66,452 | 40,125,618 | 147,868 | 65,944 |
|  | 1 | 52,593,839 | 466,101 | 281,313 | 6,759,591 | 62,973 | 37,404 | 6,941,224 | 63,692 | 38,088 |
|  | 2 | 16,687,407 | 245,595 | 163,380 | 2,037,187 | 32,509 | 21,974 | 2,191,654 | 33,340 | 22,146 |
|  | All | 342,896,251 | 1,825,960 | 959,994 | 48,775,168 | 242,787 | 125,830 | 49,258,496 | 244,900 | 126,178 |

**Table 6: ECE on the AliExpress and Huawei Browser datasets. The best result of each column is in boldface.**

| Method | AliExpress CTR | CVR | Huawei Browser CTR | CVR |
|---|---|---|---|---|
| Base | 0.005564 | 0.020246 | 0.000373 | 0.016186 |
| SIR | 0.005335 | 0.007328 | 0.000274 | **0.001906** |
| NeuCalib | 0.004628 | 0.008705 | 0.000129 | 0.002497 |
| AdaCalib | 0.004708 | 0.007529 | **0.000042** | 0.004615 |
| MCNet-None | **0.002634** | **0.005352** | 0.000147 | 0.004055 |
| MCNet-Field | 0.005195 | 0.007270 | 0.000172 | 0.002420 |

**Table 7: PCOC under every 2 training epochs (10 epochs in total).**

| Dataset | Method | PCOC (2-10 epochs) |
|---|---|---|
| AliExpress | MCNet-None | 1.0999 \| 1.1127 \| 1.1089 \| 1.1089 \| 1.1165 |
|  | MCNet-None-Aux | 1.0168 \| 0.9895 \| 0.9837 \| 0.9753 \| 0.9678 |
| Huawei Browser | MCNet-None | 0.9995 \| 0.9997 \| 0.9998 \| 1.0001 \| 1.0003 |
|  | MCNet-None-Aux | 0.9988 \| 1.0020 \| 1.0110 \| 0.9976 \| 1.0061 |

**Table 8: AUC under every 2 training epochs (10 epochs in total).**

| Dataset | Method | AUC (2-10 epochs) |
|---|---|---|
| AliExpress | MCNet-None | 0.7892 \| 0.7892 \| 0.7892 \| 0.7892 \| 0.7892 |
|  | MCNet-None-Aux | 0.7894 \| 0.7893 \| 0.7893 \| 0.7892 \| 0.7890 |
| Huawei Browser | MCNet-None | 0.8497 \| 0.8497 \| 0.8497 \| 0.8497 \| 0.8497 |
|  | MCNet-None-Aux | 0.8549 \| 0.8552 \| 0.8543 \| 0.8535 \| 0.8536 |

It shows that MCNet-None achieves the best ECE scores on both the CTR and CVR tasks of AliExpress, while MCNet-Field achieves comparable performance with the baselines. For Huawei Browser, the best results on CTR and CVR tasks are achieved by AdaCalib and SIR, respectively. Our MCNet demonstrates comparable performance with other baselines on Huawei Browser. Overall, the results of ECE are consistent with those on PCOC largely (see Table 1).

The reason might be that both ECE and PCOC do not consider the fine-grained field information.

## B.4 Model Robustness against Overfitting

MCNet is designed to be robust against overfitting via three strategies: 1) constraining the calibration function of each bin to be monotonic regarding the uncalibrated scores, 2) applying $L_2$ regularization and employing a small number of training epochs, 3) taking simple inputs, i.e., only the uncalibrated scores (MCNet-None) or together with the context features (MCNet-Field) [26]. The auxiliary neural network is an optional module of MCNet to enhance the ranking performance. To avoid overfitting, the auxiliary network is implemented as a simple 2-layer MLP. Table 7 and Table 8 report the PCOC and AUC scores on the CVR task under every 2 training epochs (10 epochs in total), demonstrating that a training epoch within the range of 2 to 10 has a negligible impact on the final calibration and ranking performance. Therefore, MCNet is robust against overfitting, even with the auxiliary network incorporated.

