# OpenReview forum: "MCNet: Monotonic Calibration Networks for Expressive Uncertainty Calibration in Online Advertising"
_ACM.org/TheWebConf/2025/Conference — WWW 2025 Poster_

### Official Review · Reviewer_R7ze · 2024-12-02

**Novelty:** 5
**Technical Quality:** 5

**Review:**

The paper introduces a novel hybrid model called MCNet, designed to improve uncertainty calibration in online advertising. MCNet addresses the limitations of existing post-hoc calibration methods by utilizing a Monotonic Calibration Function (MCF), an order-preserving regularizer, and a field-balance regularizer.

**Quality**: The paper presents the post-hoc calibration problem and offers a solution with high expressiveness, ensuring order preservation, context-awareness, and balanced performance across different data fields. The authors provide a clear and well-structured explanation of the MCNet design, supported by theoretical foundations. Experimental results validate the effectiveness of the proposed approaches on both public and private datasets, including detailed ablation studies.

**Clarity**: The paper clearly motivates the problem and highlights key challenges and design requirements for calibration in online advertising. It is well-structured, with the proposed solution thoughtfully compared against existing methods. The evaluation is divided into four sections, making it easy to follow. However, the paper does not fully address some high-level motivations about post-hoc approaches for out of domain readers, such as the root causes of miscalibration and the justification for a post-hoc strategy.

**Originality**: The paper effectively compares MCNet with existing methods. The integration of monotonic neural networks for calibration and the introduction of field-specific regularization are innovative contributions, particularly within the online advertising domain.

**Significance**: The proposed solution effectively addresses the calibration problem in online advertising with improved prediction accuracy and fairness, making it practically valuable for optimizing revenue.

**Pros and cons**:

\+ novel integration of the monotonic network and the use of regularizers

\+ comprehensive evaluation along with detailed ablation studies and comparisons with existing methods.

\+ clear presentations

\- regularization techniques and their hyperparameter tuning might pose practical challenges.

\-  monotonicity regularizers in improving calibration could be elaborated further.

\- does not adequately explain the root causes of miscalibration or why post-hoc approaches are preferred

**Questions:**

- Could you provide more details on the primary causes of miscalibration in the datasets? How did these factors influence the decision to adopt a post-hoc calibration approach rather than an alternative?

- What motivated the choice of a post-hoc calibration method over an end-to-end calibration strategy? What trade-offs or practical considerations were involved in this decision?

- Could you explain the concept of field imbalance in more detail? What are the potential causes of this imbalance—are they primarily due to differences in data distribution? From Table 1, it seems that adding the field-balance regularizer doesn’t consistently improve performance. Could you clarify under what conditions it is most effective?

- How effective is the order-preserving regularizer in maintaining monotonicity? Are there any scenarios or edge cases where it might fail to enforce the desired behavior?

- What is the rationale for using different learning rates for MCNet-None and MCNet-Field? How does this choice affect the model’s convergence or performance?

**Reviewer Confidence:**

3: The reviewer is confident but not certain that the evaluation is correct

**Scope:**

4: The work is relevant to the Web and to the track, and is of broad interest to the community

---

### Official Review · Reviewer_dLT4 · 2024-12-02

**Novelty:** 4
**Technical Quality:** 3

**Review:**

Existing calibration methods often struggle to accurately capture complex nonlinear relationships, consider contextual features, and deliver balanced performance across diverse data subsets. To overcome these challenges, this paper proposes a novel hybrid approach called MCNet (Monotonic Calibration Networks). MCNet comprises three key components: a monotonic calibration function (MCF), an order-preserving regularizer, and a field-balance regularizer. These components collectively enhance model performance and reliability in varied contexts. The method is tested on two large-scale datasets and compared against several baselines, demonstrating its effectiveness.

Strong Points

1. The paper is well-structured, with concise language and clear illustrations that effectively explain the MCNet architecture and experimental results.
2. MCNet is more expressive than existing methods and integrates context-awareness to improve calibration.
3. Extensive experiments with a range of baselines validate MCNet’s effectiveness.
4. The authors address a critical limitation: MCNet cannot be trained with traditional stochastic gradient descent (SGD) methods. They provide detailed training algorithms alongside empirical analyses of time and memory efficiency.

Weak Points
1. The model’s performance gains vary significantly across datasets, excelling on the AliExpress dataset but showing more modest improvements on others.
2. While the model demonstrates improvements in PCOC, F-RCE, and AUC metrics, the gains are not always substantial.
3. The paper lacks a discussion of potential weaknesses, future directions, or opportunities for further improvement.

**Questions:**

Please address the concern above. Furthermore, two questions:
1. Could you provide a clear definition of CVR calibration?
2. Could you explain why the auxiliary network might negatively impact the model’s calibration ability?

**Reviewer Confidence:**

3: The reviewer is confident but not certain that the evaluation is correct

**Scope:**

3: The work is somewhat relevant to the Web and to the track, and is of narrow interest to a sub-community

---

### Official Review · Reviewer_9snk · 2024-12-06

**Novelty:** 6
**Technical Quality:** 6

**Review:**

The paper presents a model for calibrating CTR and CVR prediction models. The model is based on an integral function whose parameters are optimized as neural network parameters using numerical integration methods, with a loss function incorporating two regularization terms that ensure monotonicity and balance across different fields. The paper demonstrates solid mathematical reasoning, supported by clear proofs and thorough experimental validation through comparative analysis with other models and testing on two datasets.

**Questions:**

The primary concern is the model's applicability in real-world scenarios. The training time exceeds that of NeuCalib and AdaCalib. One can reasonably assume that inference time is also longer, which conflicts with the usual demands given that calibration models are generally expected to be computationally efficient. Did the authors compare inference times with the baselines?


It seems the model's efficiency could be improved by avoiding training on a loss function with an order-preserving component. Instead, one could input items sorted by predicted CTR and calculate model scores as cumulative increments for each subsequent item. To illustrate: if a model initially outputs scores of 0.2, 0.1, and 0.5, we could input the sorted sequence (0.1, 0.2, 0.5) to the calibration model. The model could then generate an initial score (e.g., 0.08) and incremental adjustments for each element (e.g., 0.01, 0.2, 0.1), resulting in calibrated scores of 0.09 (0.08+0.01), 0.29 (0.08+0.01+0.2), and 0.39 (0.08+0.01+0.2+0.1). These scores would then be reordered according to the original item sequence, yielding calibrated predictions of 0.29, 0.09, and 0.39.

This approach could simplify the loss function, potentially facilitating easier model training and reducing the required number of parameters, thereby decreasing both training and inference times. Did the authors explore such computationally less intensive, albeit more straightforward approaches?

**Reviewer Confidence:**

3: The reviewer is confident but not certain that the evaluation is correct

**Scope:**

3: The work is somewhat relevant to the Web and to the track, and is of narrow interest to a sub-community

---

### Official Review · Reviewer_WcM8 · 2024-12-10

**Novelty:** 5
**Technical Quality:** 5

**Review:**

This paper introduce new hybrid model for improving uncertainty calibration in online advertising MCNet Monotonic Calubration Network) with special function - MCF (monotonic calibration function). Also authors present special Losses to train their network. The paper is ended with experiments on both public and industrial datasets.

Authors compared their approach for calibration function with binning-based methods, scaling-based methods and some of hybrid methods. Paper highlights weaknesses of existing approaches, which is making motivation for this paper clear.

Monotonic property is feasible in the proposed system by using motonic neural network with positive derivative (which is one of the classic approaches).

The MCNet exploit some ideas, which help MCNet to have their main properties:
1. Monotonic Calibration Function
2. Order-preserving Regularizer
3. Field-Balance Regularizer

Authors answer 4 research questions in their experiments to provide importance and perfomance of proposed method.

1. Comparing with SOTA baselines
2. Study on the auxilary NNs
3. Analysis of calibrated function
4. Analysis of hyperparametres

All in all, it a good study, which provides important analysis of properties and solutions for calibrating CTR models.

**Questions:**

Your paper is good written, but I still have some questions

1. Are you planning to publish your code with your implementation of MCNet to make your research reproducible?
2. In intoduction section you refer to the Figure 1, to find examples of binning based and hybrid methods. But you didn't draw any results for scaling-based methods. Why are they not as important as other methods?
3. I am got confused by your Figure 1 - I can find only situations, when uncalibrated score is underestimates probability. But is it common situation or just a special case of problem?
4. Lines 506-512, can you add some intuition why you are using logloss, which should be used in training function $g$? I suppose we should use it or other classification losses before we start working with calibration problem.
5. You use $c$ for field feature and in AliExpress dataset you use it as indicator of country. But can we use some smaller categories? For example electronic and household goods. I suppose, the paper will be better if you add some discussion on this parameter.
4. Can you make your pictures suitable for black&white print? For example, Figures 3,4

**Reviewer Confidence:**

2: The reviewer is willing to defend the evaluation, but it is likely that the reviewer did not understand parts of the paper

**Scope:**

4: The work is relevant to the Web and to the track, and is of broad interest to the community